# Tumor-derived CSF-1 induces the NKG2D ligand RAE-1δ on tumor-infiltrating macrophages

Thornton W Thompson, Benjamin T Jackson, P Jonathan Li, Jiaxi Wang, Alexander Byungsuk Kim, Kristen Ting Hui Huang, Lily Zhang, David H Raulet*

Department of Molecular and Cell Biology, Cancer Research Laboratory, University of California, Berkeley, Berkeley, United States

**Abstract** NKG2D is an important immunoreceptor expressed on the surface of NK cells and some T cells. NKG2D recognizes a set of ligands typically expressed on infected or transformed cells, but recent studies have also documented NKG2D ligands on subsets of host non-tumor cells in tumor-bearing animals and humans. Here we show that in transplanted tumors and genetically engineered mouse cancer models, tumor-associated macrophages are induced to express the NKG2D ligand RAE-1δ. We find that a soluble factor produced by tumor cells is responsible for macrophage RAE-1δ induction, and we identify tumor-derived colony-stimulating factor-1 (CSF-1) as necessary and sufficient for macrophage RAE-1δ induction in vitro and in vivo. Furthermore, we show that induction of RAE-1δ on macrophages by CSF-1 requires PI3K p110α kinase signaling. Thus, production of CSF-1 by tumor cells leading to activation of PI3K p110α represents a novel cellular and molecular pathway mediating NKG2D ligand expression on tumor-associated macrophages.

DOI: https://doi.org/10.7554/eLife.32919.001

*For correspondence:
raulet@berkeley.edu

## Introduction

NKG2D is a lectin-like cell surface immunoreceptor expressed on all NK cells and some T cell subsets (*Raulet, 2003*). NKG2D recognizes a diverse set of MHC-like proteins. In mice, these include the RAE-1 family (including isoforms α, β, γ, δ, and ε), the H60 family (a, b, c), and MULT1. Human NKG2D ligands include the ULBP family (with isoforms 1–6) and the MICA and MICB proteins (*Raulet et al., 2013*).

Interactions between NKG2D and its ligands mediate diverse immunological functions. Acute ligation of NKG2D on NK cells transmits a powerful activation signal through the DAP10 and DAP12 adaptors, triggering NK cell release of cytotoxic granules and pro-inflammatory cytokines such as interferon-γ (*Raulet, 2003*). In contrast, recent studies have shown that steady-state interactions of NKG2D with ligands in vivo, such as with endogenous expression of RAE-1ε in mice (*Thompson et al., 2017*) or transgenically enforced overexpression of various NKG2D ligands in vivo (*Oppenheim et al., 2005*; *Wiemann et al., 2005*), cause NK cells to adopt a state of global desensitization to acute activation. NKG2D has also been implicated as a co-stimulatory molecule for T cells (*Bauer et al., 1999*; *Markiewicz et al., 2005*), and in some cases NKG2D can mediate lymphocyte trafficking to sites of inflammation (*Markiewicz et al., 2012*). Various auto-inflammatory conditions, such as atherosclerosis in a mouse model, have also been shown to be mediated in part by NKG2D (*Ogasawara et al., 2004*; *Xia et al., 2011*; *Guerra et al., 2013*). Thus, NKG2D has diverse roles in immune cell activation and regulation depending on the cellular and physiological context.

Most cells in healthy mice lack surface NKG2D ligand expression, whereas many tumors and infected cells show expression in vitro and in vivo (*Raulet et al., 2013*). NKG2D ligand expression is

tightly regulated at multiple levels of biogenesis. In general, NKG2D ligand expression on diseased cells is usually understood as a cellular response to stresses associated with transformation or infection (*Eagle et al., 2006*; *Mistry and O'Callaghan, 2007*; *Raulet et al., 2013*). A prominent example is the induction of NKG2D ligands in mouse and human cells as a result of an activated DNA damage response (*Gasser et al., 2005*). Subsequent studies found that rapidly proliferating fibroblasts upregulate NKG2D ligands in vitro and in vivo independently of the DNA damage response, due to transactivation of the promoter of the *Raet1e* gene (which encodes RAE-1ε) by E2F transcription factors (*Jung et al., 2012*). Heat shock stress and the integrated stress response have also been implicated in NKG2D ligand expression (*Groh et al., 1996*; *Venkataraman et al., 2007*; *Nice et al., 2009*; *Gowen et al., 2015*). In some cells, steady-state expression of micro-RNAs may confer post-transcriptional regulation of NKG2D ligand expression (*Heinemann et al., 2012*; *Codo et al., 2014*). In human but not mouse cells, activation of p53 has also been implicated in NKG2D ligand induction (*Li et al., 2011*; *Textor et al., 2011*; *Iannello et al., 2013*). Thus, animals have evolved numerous mechanisms to sense abnormal cellular activity and alert the immune system through NKG2D.

Interestingly, some reports have described NKG2D ligand expression on cells that are not themselves infected or transformed. For example, Toll-like receptor (TLR) agonists induced NKG2D ligands on mouse macrophages and human monocyte-derived dendritic cells (*Hamerman et al., 2004*; *Ebihara et al., 2007*). There is also increasing evidence that subsets of tumor-associated cells show NKG2D ligand induction in animals and humans. Tumor-associated myeloid cells and circulating monocytes in glioblastoma patients were shown to upregulate NKG2D ligands (*Crane et al., 2014*). In transplant and spontaneous mouse models, tumor-associated endothelial cells were found to induce high levels of the NKG2D ligand RAE-1ε (*Thompson et al., 2017*). Expression of RAE-1 molecules was also found on macrophages infiltrating a mouse model of melanoma and a model of lymphoma (*Deng et al., 2015*; *Nausch et al., 2008*).

Tumors establish a complex microenvironment characterized by an intricate interplay between cancer cells and associated stroma. Some tumor-infiltrating cells, such as cytotoxic lymphocytes, can be activated to kill tumor cells and protect the host (*Vesely et al., 2011*). Other tumor-associated stroma can have pleiotropic effects depending on tumor type and physiological context. For example, many tumors are extensively infiltrated by macrophages, which often have pro-tumor functions such as promoting angiogenesis or impairing the functions of cytotoxic lymphocytes, but can also exert anti-tumor activities depending on the molecular and cellular milieu (*Noy and Pollard, 2014*). Macrophages can sense the character of tumor microenvironments using an array of receptors and respond to different microenvironments by expressing various secreted and surface-bound immunomodulatory molecules (*Noy and Pollard, 2014*). Understanding the cellular and molecular factors that control the activity and expression profile of tumor-associated macrophages is critical to understanding tumor microenvironments and revealing new targets for therapy.

Here we show that the NKG2D ligand RAE-1δ is induced on tumor-associated macrophages but not other cells that infiltrate several models of transplanted and autochthonous cancer. Unexpectedly, we find that the cytokine colony-stimulating factor-1 (CSF-1) is released by tumor cells and is necessary and sufficient to induce RAE-1δ at the mRNA and cell surface levels on macrophages in vitro and on tumor-associated macrophages in vivo. Furthermore, we show that the p110α catalytic subunit of PI3K is required for CSF-1-mediated macrophage RAE-1δ induction. Thus, tumor cell secretion of CSF-1 is sensed by macrophages through CSF-1R and PI3K p110α, leading to induction of the NKG2D ligand RAE-1δ.

## Results

### RAE-1δ induction on tumor-associated macrophages

A limited number of studies have described NKG2D ligand expression on subsets of tumor-associated hematopoietic cells (*Crane et al., 2014*; *Deng et al., 2015*; *Nausch et al., 2008*). To further investigate this phenomenon, we used flow cytometry to analyze NKG2D ligands on hematopoietic cells infiltrating several transplant tumor models. First, WT C57BL/6 mice were injected subcutaneously with a high dose ($1 \times 10^6$) of B16-BL6 melanoma cells, hereafter referred to as B16. Once established at approximately 1 cm in diameter (10–17 days post-injection), tumors were dissociated and stained with lineage markers and monoclonal antibodies for NKG2D ligands, including RAE-1δ,

RAE-1ε, MULT1, or a polyclonal antibody that recognizes multiple H60 isoforms. As RAE-1 molecules are quite similar, we validated the specificity of the antibodies by staining B16 cells transduced with RAE-1δ or RAE-1ε with the antibodies targeting these ligands (*Figure 1—figure supplement 2A*), and we previously confirmed isoform-specific blocking by these antibodies (*Thompson et al., 2017*).

Tumor-associated macrophages (hereafter called TAMs) are an important subset of myeloid cells identified as CD45-pos; CD11b-hi; Ly6G-neg; F4/80-hi (*Figure 1—figure supplement 1A*). Interestingly, TAMs in B16 tumors expressed RAE-1δ but not other NKG2D ligands (*Figure 1A*). In addition to strong expression on TAMs, RAE-1δ was weakly expressed on monocytes in B16 tumors (identified as CD45-pos; CD11b-hi; Ly6G-neg; F4/80-low; Ly6C-hi – gating strategy in *Figure 1—figure supplement 3A*) – but negligible on other hematopoietic cells (*Figure 1B*). Importantly, RAE-1δ staining on TAMs was completely absent in RAE-1-KO mice, which contain frameshift mutations in the genes encoding RAE-1δ and RAE-1ε, confirming the specificity of the RAE-1δ staining (*Figure 1—figure supplement 1B*). In contrast to robust TAM expression of RAE-1δ, splenic macrophages, peritoneal macrophages, and blood monocytes in mice bearing B16 tumors expressed little to no RAE-1δ (*Figure 1—figure supplements 1C and 3B,C*). These data indicate that macrophages within the B16 tumor microenvironment are induced to express the NKG2D ligand RAE-1δ. Expression of RAE-1δ in TAMs within B16 tumors was similar at various stages of tumor growth (*Figure 1—figure supplement 2B*). Gating strategies for blood monocytes and peritoneal macrophages are shown in *Figure 1—figure supplement 3*.

In contrast to the findings with B16 tumors, RAE-1δ staining was negligible or very low on TAMs in similarly sized S.C. tumors generated by injection of the RMA-S T cell lymphoma cell line ($5 \times 10^6$ cells injected) (*Figure 1C*). We next sought to analyze NKG2D ligands on tumor-associated cells in spontaneous tumor models. In the KP sarcoma model driven by lentiviral-Cre activation of oncogenic *Kras* and deletion of *Trp53* (*DuPage et al., 2009*), TAMs in primary tumors expressed robust RAE-1δ (*Figure 1C*). In contrast, TAMs within primary TRAMP prostate tumors – a spontaneous adenocarcinoma model driven by expression of SV40 T antigens (*Greenberg et al., 1995*) – mostly lacked RAE-1δ (*Figure 1C*). Together, these data indicate that TAMs, but not other hematopoietic cells, are induced to express RAE-1δ in some transplant and spontaneous tumors (B16 tumors and primary KP sarcomas), but not in others (RMA-S tumors and primary TRAMP adenocarcinomas) (see *Figure 1—figure supplement 2C* for comparisons). Thus, tumor microenvironments are differentially capable of inducing NKG2D ligand expression by macrophages.

## A tumor-derived soluble factor induces RAE-1δ on macrophages in vitro

To interrogate the mechanism of RAE-1δ induction on TAMs, we began by testing the hypothesis that a soluble factor released from tumor cells induces macrophage RAE-1δ. Resident macrophages were obtained from naïve WT mice by peritoneal lavage and cultured ex vivo with concentrated cell culture medium from B16 cells (diluted 1:1 with fresh medium) or similarly diluted concentrated fresh medium as a control. Macrophages cultured in the control medium showed little to no RAE-1δ expression, but culture with B16-conditioned medium led to a robust induction of cell surface RAE-1δ (*Figure 2A*). RAE-1δ was similarly induced by culture medium from a KP sarcoma cell line (*Figure 2B*). These results indicated that a soluble factor(s) produced by B16 tumor cells and KP sarcoma cells is sufficient to induce RAE-1δ on macrophages ex vivo.

## CSF-1 is sufficient to induce macrophage RAE-1δ ex vivo

To identify soluble factors that induce RAE-1δ on macrophages, we stimulated peritoneal macrophages with a panel of recombinant cytokines known to ligate receptors expressed on macrophages (*Table 1*). Alone among the cytokines tested, recombinant colony-stimulating factor-1 (CSF-1), also known as macrophage colony-stimulating factor (MCSF), was sufficient to induce robust RAE-1δ expression on macrophages (*Figure 2C*). Macrophages express the CSF-1 receptor (CSF-1R) (*Figure 2—figure supplement 1A*), and macrophages cultured with recombinant CSF-1 along with blocking antibody against CSF-1R failed to induce RAE-1δ, establishing that the added cytokine acts through CSF-1R (*Figure 2C*). We performed qPCR on reverse-transcribed RNA from CSF-1-stimulated macrophages and found that recombinant CSF-1 caused upregulation of transcripts of the *Raet1d* gene, which encode RAE-RAE-1δ (*Figure 2D*). Stimulation with graded doses of CSF-1 showed that as little as 3 ng/ml was sufficient to induce detectable RAE-1δ in this system, with high

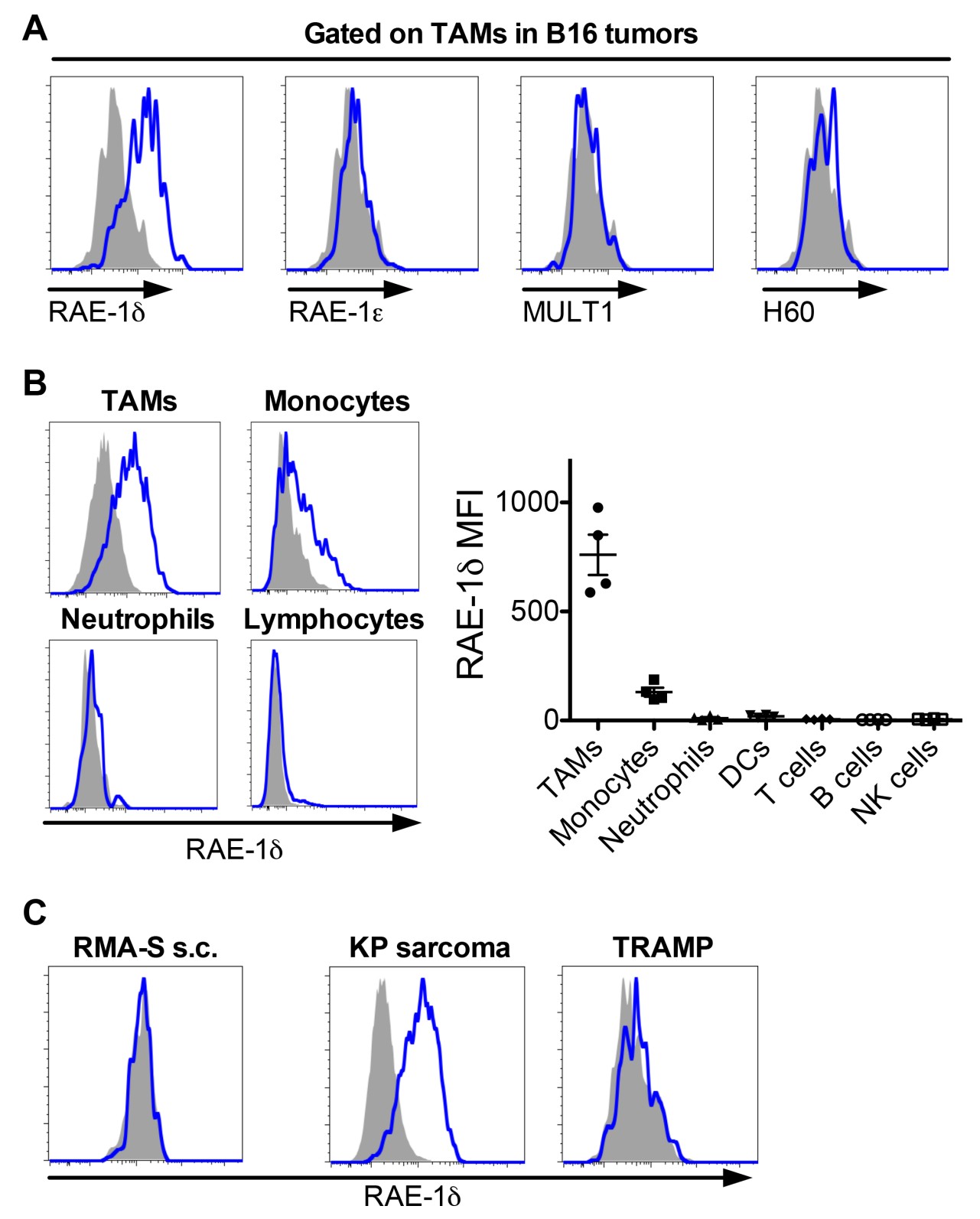

**Figure 1.** RAE-1δ is induced on tumor-associated macrophages in subcutaneously transferred and spontaneous tumors. (**A**) Established B16 S.C. tumors were dissociated and analyzed for NKG2D ligand expression on tumor-associated macrophages. (**B**) RAE-1δ expression (left) and MFI quantification (right) on the indicated cell types in B16 tumors. (**C**) RAE-1δ expression on TAMs in spontaneous KP sarcoma, but not in spontaneous TRAMP prostate adenocarcinoma or transferred RMA-S lymphoma. Data are representative of >3 independent experiments.

*Figure 1 continued on next page*

*Figure 1 continued*

DOI: https://doi.org/10.7554/eLife.32919.002

The following figure supplements are available for figure 1:

**Figure supplement 1.** Gating strategy and RAE1δ expression on tumor-associated macrophages and monocytes in mice with B16 tumors.

DOI: https://doi.org/10.7554/eLife.32919.003

**Figure supplement 2.** RAE-1 antibody validation and RAE-1δ staining on TAMs in different tumors.

DOI: https://doi.org/10.7554/eLife.32919.004

**Figure supplement 3.** Gating strategies for blood and tumor-associated monocytes and peritoneal macrophages.

DOI: https://doi.org/10.7554/eLife.32919.005

induction levels seen at 10 ng/ml (*Figure 2—figure supplement 1B*). Interestingly, induction of other NKG2D ligands by CSF-1 was negligible (*Figure 2E*), indicating that CSF-1 upregulates RAE-1δ highly selectively. Macrophages can be derived from bone marrow cells in vitro using CSF-1 or GM-CSF. Consistent with our findings, macrophages derived from bone marrow cells via 7 days of culture with CSF-1 induced robust RAE-1δ, whereas parallel cultures in GM-CSF showed little to no RAE-1δ expression (*Figure 2—figure supplement 1C*)

## CSF-1 is necessary for macrophage RAE-1δ induction by tumor cell supernatants ex vivo

To further assess whether CSF-1 contributes to induction by tumor cells of RAE-1δ on macrophages, we analyzed CSF-1 secretion by B16 cells (in which TAMs express RAE-1δ – *Figure 1A*) and RMA-S cells (in which TAMs lack RAE-1δ – *Figure 1C*). As measured by ELISA of cell culture supernatants, B16 cells secreted substantial CSF-1, whereas RMA-S cells did not (*Figure 3A*). KP sarcoma cell lines also produced CSF-1, and much more robustly than did B16 cells (*Figure 3—figure supplement 1A*). We used ELISA to analyze CSF-1 protein levels in tumor microenvironments in vivo from mechanically dissociated S.C. tumors and found the concentrations of intratumoral CSF-1 were much greater in B16 tumors than in RMA-S tumors (*Figure 3B*). Furthermore, the level of CSF-1 within B16 tumors was substantially greater than serum CSF-1 levels in naïve or tumor-bearing mice (*Figure 3—figure supplement 1B*), consistent with previous reports describing steady-state CSF-1 levels in circulation (*Menke et al., 2009*).

These observations suggested that tumor cell secretion of CSF-1 might contribute to macrophage RAE-1δ induction. To directly test this hypothesis in vitro, peritoneal macrophages were cultured with concentrated B16-conditioned medium in the presence of control Ig or anti-CSF-1R blocking antibody. We found that CSF-1R blockade completely abrogated macrophage RAE-1δ induction by B16-conditioned medium (*Figure 3C*). RAE-1δ induction by KP cell line-conditioned medium was also completely prevented by antibody blockade of CSF-1R (*Figure 3D*). Collectively, these data indicated that CSF-1 is sufficient to induce RAE-1δ on macrophages, and that CSF-1 is necessary for macrophage RAE-1δ induction by B16 and KP tumor cell supernatants in vitro.

## Short-term blockade of CSF-1 or CSF-1R abrogates TAM RAE-1δ expression in vivo

We sought to determine whether the CSF-1/CSF-1R axis controlled RAE-1δ expression on TAMs in vivo. Mice with established B16 tumors were treated with anti-CSF-1 or anti-CSF-1R, and RAE-1δ on TAMs was analyzed 48 hr post-treatment. Blockade of CSF-1 or CSF-1R each led to substantial reductions in RAE-1δ expression by TAMs (*Figure 4A*). As it has been shown that steady-state CSF-1 signaling is necessary for monocyte and macrophage survival in vivo, we injected tumor-bearing mice with CSF-1R antibody and monitored tumor-infiltrating macrophage numbers and RAE-1δ expression at various time points. Blockade of CSF-1R for 2 days had no impact on macrophage cell numbers but drastically reduced macrophage RAE-1δ expression, whereas treatments for 5 days or longer caused a major depletion in TAM numbers, associated with low RAE-1δ levels on the few remaining macrophages (*Figure 4—figure supplement 1A*). Similar to the findings with B16 tumors, a 2 day treatment with CSF-1R antibody resulted in a substantial reduction in RAE-1δ on TAMs in primary KP tumors without a significant reduction in TAM numbers, generalizing our findings to spontaneous tumors (*Figure 4B* and *Figure 4—figure supplement 1B*). Thus, antibody blockade of the

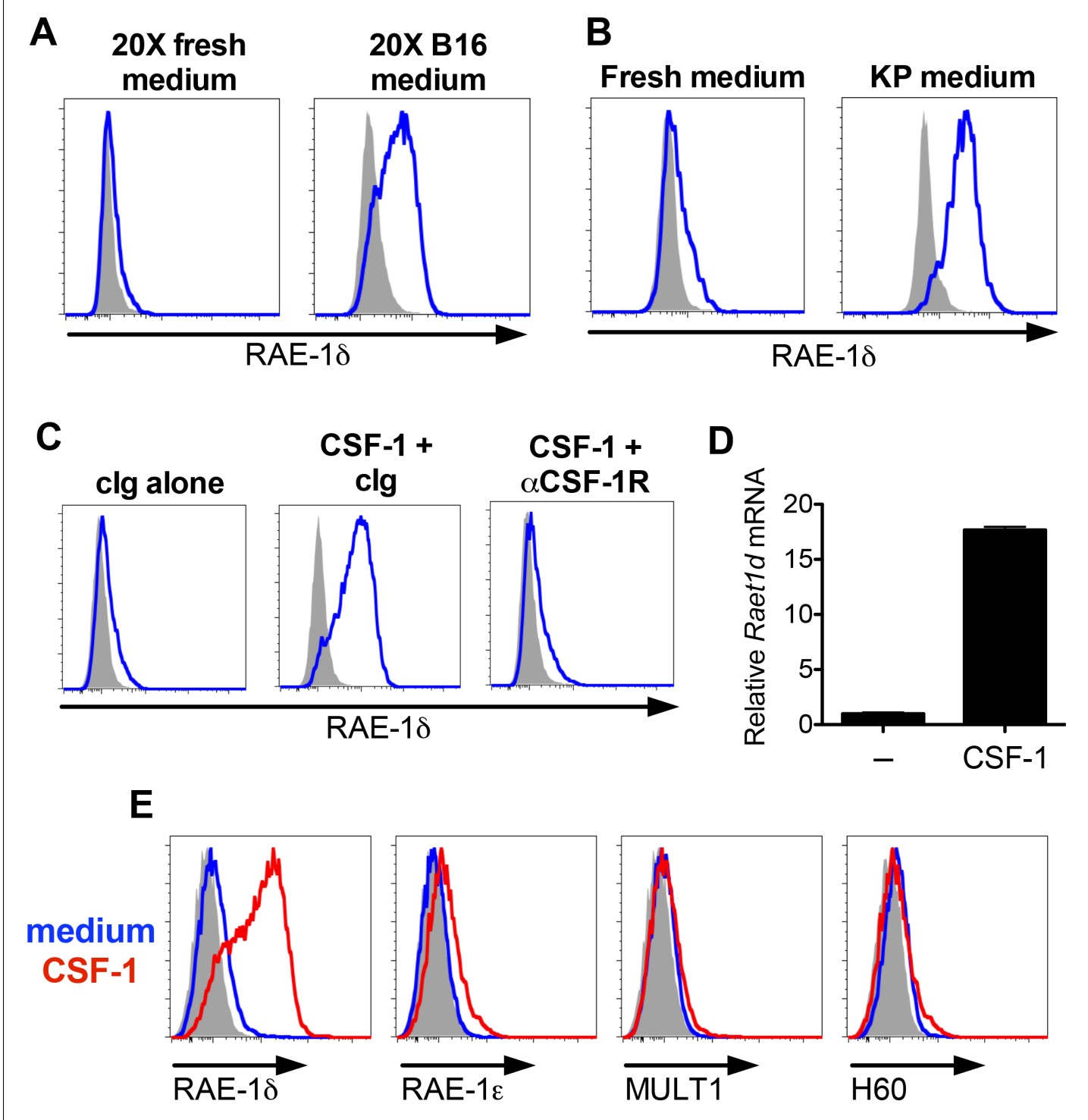

**Figure 2.** B16 and KP cell line conditioned medium and CSF-1 induces RAE-1δ on macrophages. (A) Peritoneal wash cells were cultured with a 1:1 mixture of fresh medium plus 20X concentrated fresh medium or 20X concentrated B16 cell culture supernatants, and macrophage RAE-1δ was analyzed by flow cytometry 48 hr later. (B) Peritoneal wash cells were stimulated 48 hr ex vivo with a 1:1 mixture of fresh medium supplemented with fresh medium or conditioned medium from cultures of a KP sarcoma cell line generated from a primary KP sarcoma, and macrophage RAE-1δ was analyzed 48 hr later by flow cytometry. (C) Peritoneal wash cells were cultured with or without 10 ng/ml CSF-1, with the addition of control Ig or CSF-1R antibody (1 μg/ml), and macrophage RAE-1δ was analyzed 48 hr later by flow cytometry. (D) Peritoneal macrophage *Raet1d* mRNA 48 hr after stimulation with or without the addition of CSF-1 (10 ng/ml). (E) Peritoneal macrophage expression of the indicated NKG2D ligands 48 hr after stimulation with CSF-1 or control medium. Data are representative of >3 independent experiments.

*Figure 2 continued on next page*

*Figure 2 continued*

DOI: https://doi.org/10.7554/eLife.32919.006

The following figure supplement is available for figure 2:

**Figure supplement 1.** Peritoneal macrophage CSFR1 expression and dose-dependent RAE-1δ induction by CSF-1, and bone marrow macrophage stimulation with CSF-1 or GM-CSF.

DOI: https://doi.org/10.7554/eLife.32919.007

CSF-1/CSF-1R axis suppresses RAE-1δ expression by TAMs in B16 S.C. tumors and autochthonous KP sarcomas.

## Tumor-derived CSF-1 is required for RAE-1δ expression by TAMs in vivo

To formally test whether tumor-derived CSF-1 was responsible for inducing TAM RAE-1δ, we used CRISPR/Cas9 to target the *Csf1* open reading frame for deletion in B16 cells. B16 cells were transiently transfected with plasmids encoding Cas9 and two guide RNAs targeting loci immediately adjacent to the *Csf1* ORF. Transfected cells were single-cell cloned, and clones were analyzed for CSF-1 secretion by ELISA. CSF-1-negative cells were injected into WT mice alongside parental B16 tumors, and established tumors were analyzed for RAE-1δ expression by TAMs. Mice were given a high dose ($1 \times 10^6$ cells) to standardize tumor growth rates. Compared with control tumors, *Csf1*-KO B16 tumors showed markedly lower RAE-1δ expression by TAMs (*Figure 5A*). Mice injected with a second, independent *Csf1*-KO B16 clone also showed substantially reduced RAE-1δ expression by TAMs (*Figure 5B*). To control for off-target effects of Cas9, *Csf1*-KO B16 cells were stably transduced with control empty vector or a *Csf1*-expression vector, and injected into mice. *Csf1*-transduction completely reversed the KO phenotype, and restored RAE-1δ expression on TAMs (*Figure 5C*).

Unlike B16 cells, RMA-S cells fail to secrete CSF-1 (*Figure 3A*) and also fail to induce significant RAE-1δ expression by TAMs (*Figure 1C*). In contrast, RMA-S cells stably transduced with a CSF-1-expression vector efficiently induced RAE-1δ expression by TAMs whereas transduction with empty vector had little or no effect (*Figure 5D*). Collectively, these data provide decisive evidence that production of CSF-1 by tumor cells in vivo drives RAE-1δ expression on tumor-associated macrophages.

## PI3K p110α signals are required for macrophage RAE-1δ induction by CSF-1

CSF-1 binds the CSF-1 receptor to initiate a variety of intracellular signaling pathways. PI3K is an important signaling molecule and a known target downstream of CSF-1R. PI3K signals have also been linked to induction of RAE-1 molecules in other contexts (*Tokuyama et al., 2011*), so we sought to determine whether PI3K activation by CSF-1 was involved in macrophage RAE-1δ induction. First, we analyzed activation of the PI3K pathway by intracellular flow cytometry for phosphorylated-S6, a known downstream target of PI3K signaling, in peritoneal macrophages stimulated with CSF-1. Indeed, macrophages showed robust S6 phosphorylation after CSF-1 stimulation (*Figure 6—*

**Table 1.** Cytokine stimulation of macrophages for RAE-1δ induction.

| Treatment | Macrophage RAE-1δ induction? |
| --- | --- |
| IL-1α | – |
| IL-1β | – |
| IL-4 | – |
| IL-6 | – |
| IL-12 | – |
| IFNβ | – |
| IFNγ | – |
| TNFα | – |
| CSF1 | +++ |

DOI: https://doi.org/10.7554/eLife.32919.008

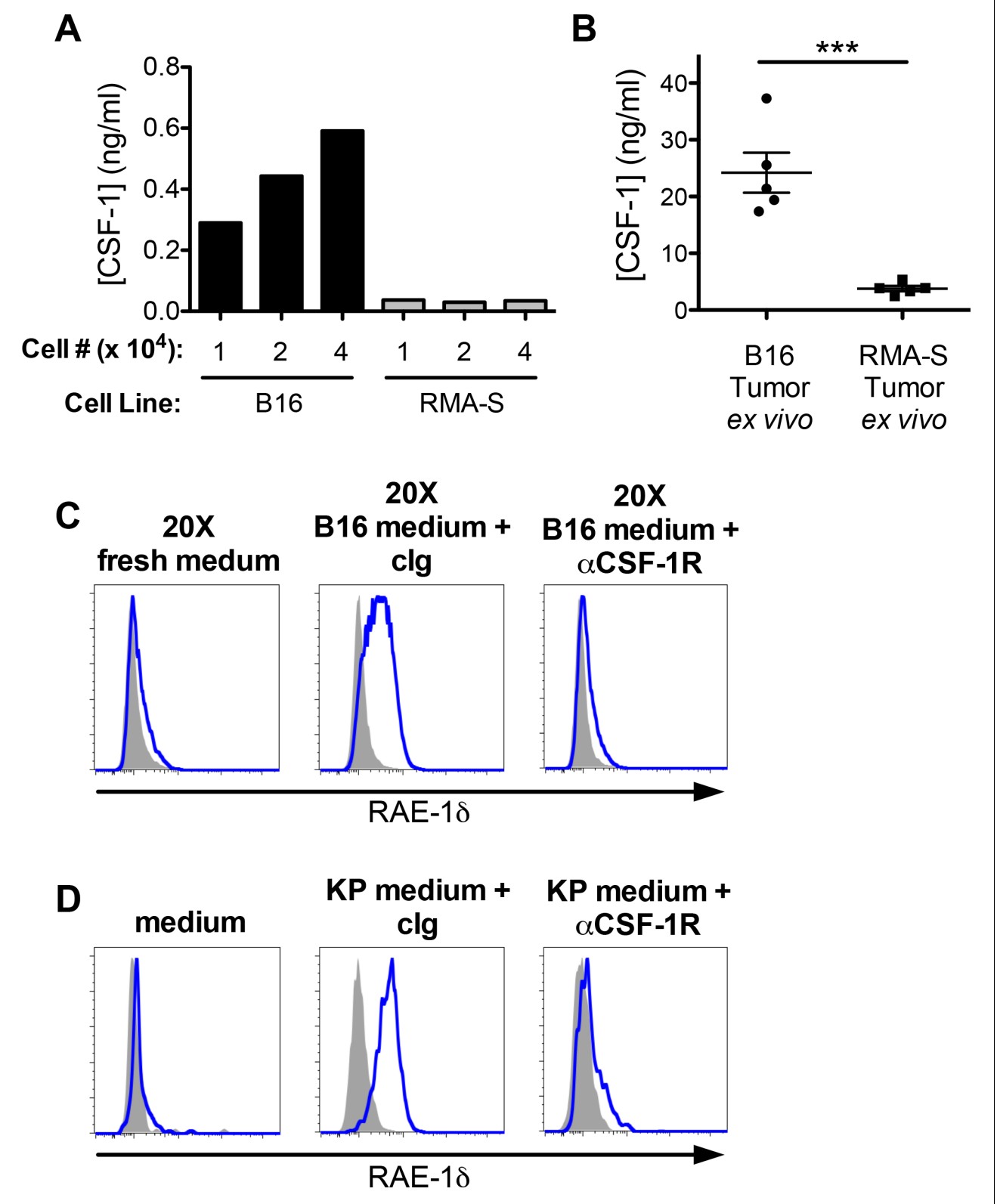

**Figure 3.** CSF-1 is necessary for macrophage RAE-1δ induction by tumor conditioned media. (A) The indicated numbers of B16 or RMA-S cells were seeded in 12-well plates, and CSF-1 levels in the supernatants were measured by ELISA 48 hr later. (B) Established B16 or RMA-S tumors were dissociated, and CSF-1 levels in dissociation supernatants were measured by ELISA; intra-tumoral concentrations were calculated using tumor volume measurements (total ng of CSF-1 divided by the tumor volume at time of harvest). (C) Peritoneal macrophage RAE-1δ expression 48 hr after culture with

*Figure 3 continued on next page*

*Figure 3 continued*

concentrated fresh medium, concentrated B16 conditioned medium plus control Ig (1 ug/ml), or concentrated B16 conditioned medium plus anti-CSF-1R (1 ug/ml). (D) Peritoneal macrophage RAE-1δ 48 hr after culture with fresh medium, KP conditioned medium plus control Ig, or KP conditioned medium plus anti-CSF-1R (1 ug/ml). Data are representative of >3 independent experiments.

DOI: https://doi.org/10.7554/eLife.32919.009

The following figure supplement is available for figure 3:

**Figure supplement 1.** In vitro and in vivo CSF-1 production in tumors.

DOI: https://doi.org/10.7554/eLife.32919.010

*figure supplement 1A*). There are four isoforms of the catalytic p110 unit of PI3K, denoted α, β, γ, and δ. We cultured macrophages with CSF-1 plus isoform-specific PI3K inhibitors and analyzed RAE-1δ induction. Interestingly, specific inhibition of PI3K p110α with two different chemical inhibitors (PI3Ka2i and PI-103) prevented CSF-1-induced RAE-1δ expression at low inhibitor concentrations, whereas inhibitors of p110 isoforms β, δ, and γ only inhibited RAE-1δ induction at high inhibitor concentrations, likely due to nonspecific inhibition or off-target effects (*Figure 6A and B*). The two PI3K p110α inhibitors also inhibited accumulation of *Raet1d* mRNA in CSF-1-treated macrophages (*Figure 6C*). These results indicate that PI3K p110α activity is required for induction of RAE-1δ gene expression by CSF-1.

## Interactions of macrophage RAE-1δ with NKG2D and NK cells

RAE-1 molecules ligate NKG2D to regulate NK cell activity. We sought to investigate the functional role of CSF-1-induced macrophage RAE-1δ using an in vitro co-culture system. Peritoneal macrophages from WT or RAE-1-KO mice were stimulated with CSF-1 for 48 hr, followed by co-culture with WT splenocytes for 18 hr. In parallel, WT splenocytes were co-cultured with B16 or B16-RAE-1δ cells for 18 hr to analyze the effect of tumor-expressed RAE-1δ vs. macrophage-expressed RAE-1δ. Following co-culture, NKG2D levels on NK cells were analyzed by flow cytometry as a measure of receptor engagement (because NKG2D is internalized from the cell surface upon engagement), or the co-cultured cells were subjected to 5 hr stimulation with platebound antibodies ligating the NK cell activating receptor NKp46, and NK cell degranulation and IFNγ expression were analyzed by flow cytometry.

Co-culture of NK cells with CSF-1-induced macrophages expressing RAE-1 molecules efficiently downregulated NKG2D from the NK cell surface, whereas co-culture with RAE-1-KO macrophages had little to no effect on NKG2D levels (*Figure 7A*). Similarly, co-culture with RAE-1δ-expressing B16 cells led to NKG2D downregulation, whereas parental B16 cells had little to no effect (*Figure 7B*). Thus, CSF-1-induced RAE-1δ on macrophages is capable of binding and engaging NKG2D, leading to receptor internalization.

NK cells co-cultured with RAE-1δ-expressing macrophages showed an augmented functional response in vitro to anti-NKp46 stimulation compared to NK cells co-cultured with RAE-1-KO macrophages (*Figure 7C*). Similarly, NK cells co-cultured with B16-RAE-1δ cells showed augmented functional responses compared with NK cells co-cultured with parental B16 cells (*Figure 7D*). These data indicate that, in this in vitro system, short-term interactions with CSF-1-stimulated macrophages expressing RAE-1δ had the effect of priming NK cells to respond better when stimulated through a distinct activating receptor. A recent report from our group explored the role of host RAE-1 molecules on NK cell function in vivo in greater detail, and is described in the discussion section.

We also considered the hypothesis that RAE-1δ-expressing TAMs were being targeted for killing in vivo. However, the frequency of TAMs among CD45+ cells in B16 tumors was similar in WT and RAE-1-KO mice (*Figure 7—figure supplement 1A*). To test whether NKG2D-RAE-1δ interactions select against TAMs with high RAE-1δ expression, we analyzed RAE-1δ expression on TAMs in B16 tumors in WT and NKG2D-KO mice. RAE-1δ levels on TAMs were similar in these two genotypes (*Figure 7—figure supplement 1B*).

## Discussion

Here we describe a novel cellular and molecular axis regulating NKG2D ligand expression on tumor-associated macrophages. We find that, in a set of transplant and spontaneous cancer models, tumor

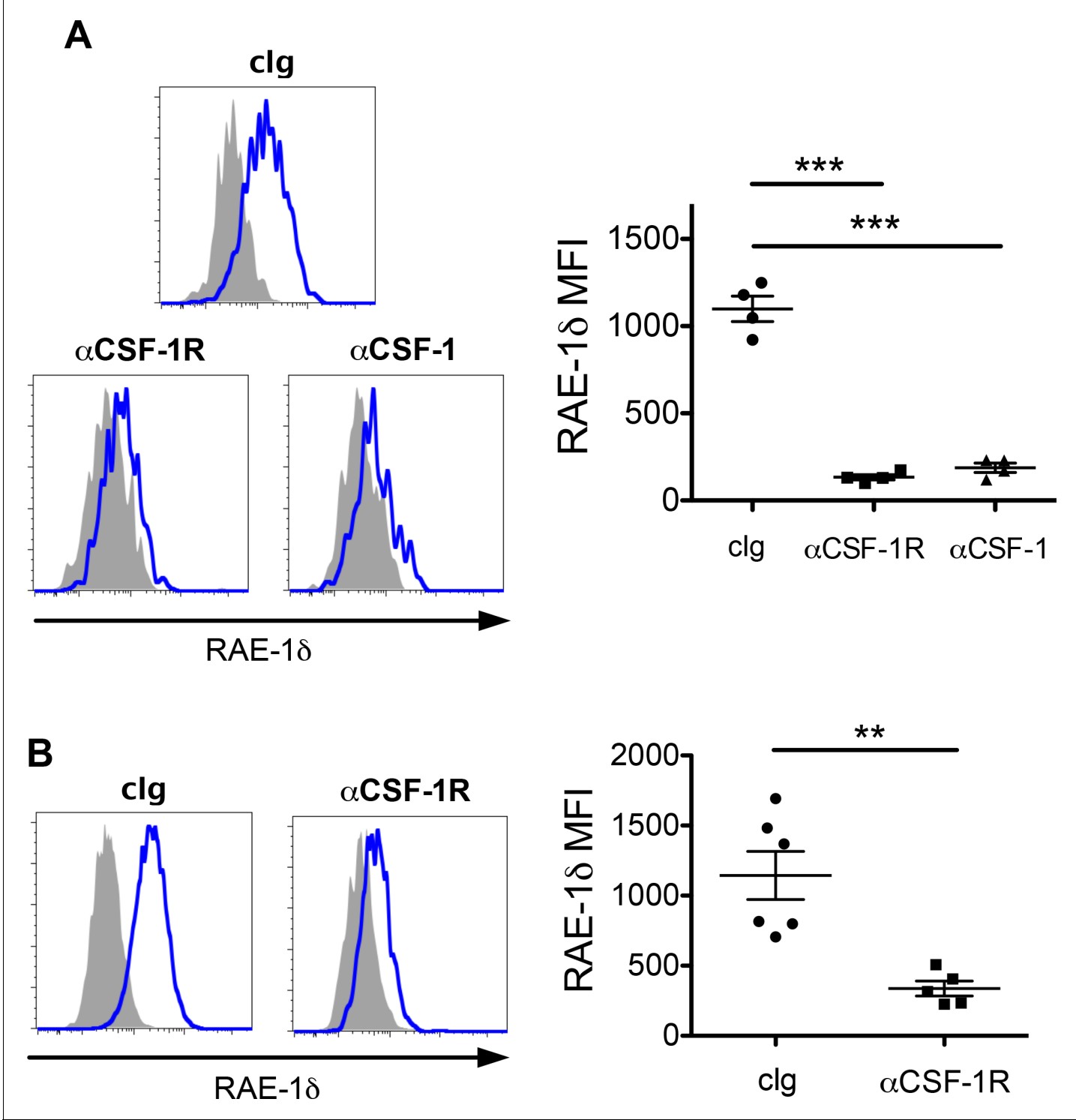

**Figure 4.** Blockade of CSF-1 or CSF-1R abrogates RAE-1δ expression by TAMs in vivo. (**A**) Mice with established B16 tumors were injected i.p. with 200 ug of the indicated antibody, and RAE-1δ on TAMs was analyzed 48 hr later. (**B**) KP mice with established sarcomas were injected i.p. with 200 ug of the indicated antibody, and RAE-1δ on TAMs was analyzed 48 hr later. Statistical significance was determined using one-way ANOVA with Bonferroni post-tests (**A**) or a two-tailed unpaired Student's t test (**B**). Data represent means ±SEM. Data are representative of >3 independent experiments.

DOI: https://doi.org/10.7554/eLife.32919.011

The following figure supplement is available for figure 4:

**Figure supplement 1.** Tumor associated macrophage numbers and RAE-1δ expression after treatments with anti-CSF-1R.

*Figure 4 continued on next page*

*Figure 4 continued*

DOI: https://doi.org/10.7554/eLife.32919.012

cells secrete the cytokine CSF-1, which induces the NKG2D ligand RAE-1δ on tumor-associated macrophages via a CSF-1R-PI3Kα-dependent signaling pathway. These data expand our knowledge of NKG2D ligand regulation on tumor-associated cells and enhance our understanding of the complex cellular and molecular dynamics within tumor microenvironments.

The specificity of NKG2D ligand induction on tumor-associated hematopoietic cells was striking. Macrophages and monocytes were the only hematopoietic cells found to express NKG2D ligands, and ligand expression on those cells was completely limited to RAE-1δ. Along with our previous study describing RAE-1 molecules expressed on tumor-associated endothelium (*Thompson et al., 2017*), these datasets represent a substantial addition to our understanding of NKG2D ligand expression on tumor-associated cells in multiple tumor models.

It is notable that macrophages treated with CSF-1 upregulated RAE-1δ but not other NKG2D ligands, matching the specificity of RAE-1δ induction on TAMs. These data suggest that regulation of NKG2D ligands is tightly controlled and highly specific, which has also been noted in other cellular and molecular contexts. We show here that PI3K p110α is required for macrophage RAE-1δ by CSF-1. A previous study also implicated PI3K p110α in NKG2D ligand induction upon in vitro MCMV infection (*Tokuyama et al., 2011*). More research is needed to ascertain whether PI3K collaborates with other signals to induce NKG2D ligands, and to uncover the downstream events linking PI3K activity to transcription of NKG2D ligands. The regulatory factors mediating this specificity remain an area of ongoing investigation.

To our knowledge, this report is the first to describe CSF-1 as an inducer of NKG2D ligands. CSF-1, but not other tested cytokines, induced RAE-1δ on macrophages. Previous reports of NKG2D ligand expression on myeloid cells are relatively limited. Stimulation of macrophages through TLRs was found to induce RAE-1 molecules but not H60 family ligands or MULT1 (*Hamerman et al., 2004*). Tumor-associated myeloid cells and circulating monocytes in glioblastoma patients were found to upregulate the human NKG2D ligands ULBP1 and MICB (*Crane et al., 2014*). A handful of reports have indicated that dendritic cells can upregulate NKG2D ligands in vitro as a result of certain maturation conditions (*Schrama et al., 2006*; *Ebihara et al., 2007*). Thus, the data presented in this manuscript represent a novel molecular signature linked to NKG2D ligand induction. Notably, these data do not implicate RAE-1δ expression within the classical (*Italiani and Boraschi, 2014*) – and controversial (*Martinez and Gordon, 2014*) – M1/M2 paradigm of macrophage activation. CSF-1 is associated with macrophage renewal and activation, but it induces a transcriptional signature distinct from M1 or M2 expression profiles (*Hume and MacDonald, 2012*). Indeed, classical M1 and M2 cytokines (IFNγ and IL-4, respectively) both failed to induce macrophage RAE-1δ in our hands (*Table 1* and not shown).

We find here that induction RAE-1δ on TAMs occurred in tumor microenvironments containing high levels of CSF-1. Steady state amounts of CSF-1 are known to be present and required for maintaining monocytes and macrophages, but those concentrations were insufficient to induce RAE-1δ expression. Hence, RAE-1δ induction requires a higher dose of CSF-1 than is necessary for survival and renewal of macrophages. These considerations suggest the intriguing possibility that macrophage detection of elevated CSF-1 levels, in this case driven by dense concentrations of CSF-1-producing tumor cells, could be considered a sensing mechanism for local disturbances in homeostasis. CSF-1 is associated with macrophage programs that promote tumor growth (*Hume and MacDonald, 2012*), and tumor cell secretion of CSF-1 may be an adaptive feature of some cancers. It is interesting to speculate on other scenarios that might resemble these CSF-1-producing tumor microenvironments.

The physiological role of macrophage RAE-1δ expression in tumors and other contexts remains enigmatic. NKG2D interactions with RAE-1 molecules are typically thought to provide an activating signal to NKG2D-expressing cells. Indeed, in the current study, macrophage RAE-1δ had a stimulatory effect in an in vitro co-culture model, as short-term incubation of CSF-1-stimulated RAE-1δ-expressing macrophages with NK cells caused elevated NK responses to subsequent acute stimulation compared to incubation with macrophages lacking RAE-1 molecules. These findings are similar

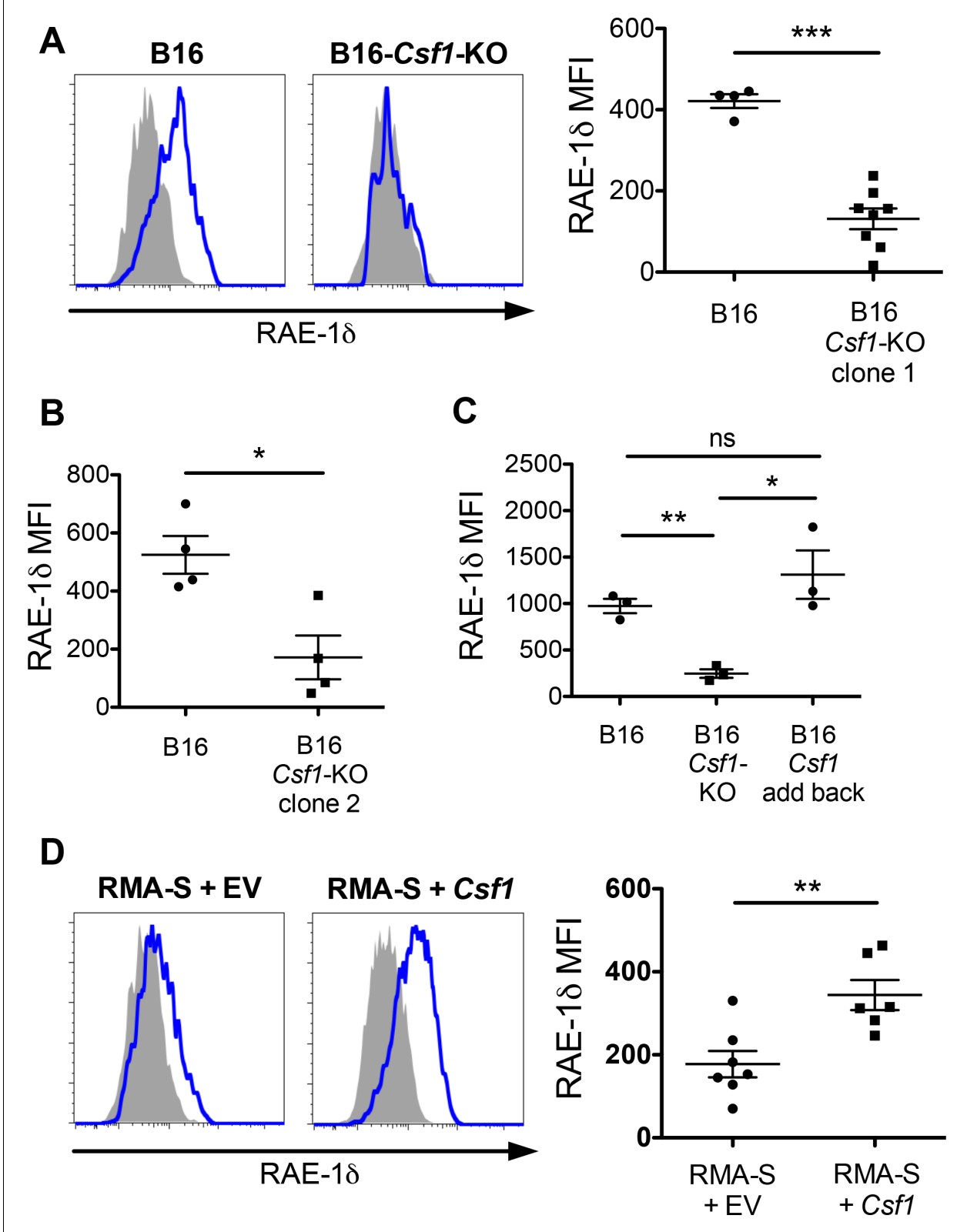

**Figure 5.** Tumor-derived CSF-1 is required for TAM RAE-1δ expression in vivo. (**A**) RAE-1δ expression on TAMs in established B16 or B16-*Csf1*-KO tumors. (**B**) RAE-1δ on TAMs in mice with established B16 tumors or tumors of a second clone of B16-*Csf1*-KO cells. (**C**) RAE-1δ on TAMs in mice with established B16, B16 *Csf1*-KO, or B16 *Csf1*-KO tumors in which CSF-1 expression had been restored by transduction (add-back tumors). (**D**) RAE-1δ on TAMs in mice with established RMA-S or RMA-S-*Csf1*-overexpressing tumors. Statistical significance was determined using one-way ANOVA with

*Figure 5 continued on next page*

*Figure 5 continued*

Bonferroni post-tests (C) or a two-tailed unpaired Student t test (A, B, D). Data represent means ±SEM, and are representative of 2–4 independent experiments.

DOI: https://doi.org/10.7554/eLife.32919.013

to a previous study showing that expression of RAE-1 molecules on mononuclear myeloid-derived suppressor cells in certain tumor-bearing mice has a stimulatory effect on NK cells in a co-culture setting (*Nausch et al., 2008*). However, a recent study from our laboratory employed antibody blockade and genetic deletion studies to show that endogenously expressed RAE-1ε ligated NKG2D and desensitized NK cells in steady state conditions and in tumors, whereas RAE-1δ had little or no effect on NKG2D levels or desensitization (*Thompson et al., 2017*). Furthermore, bone marrow chimera experiments in that study demonstrated that the impact of RAE-1 expressed by hematopoietic cells was relatively modest (*Thompson et al., 2017*). Because NKG2D is known to induce killing of RAE-1-expressing cells by NK cells in vivo, we tested the hypothesis that RAE-1δ expression may render TAMs sensitive to NKG2D-mediated killing in tumors or select for macrophages with lower RAE-1δ surface levels. However, WT and RAE-1-KO mice had similar macrophage numbers in B16 tumors, and TAM RAE-1δ levels were similar in WT and NKG2D-KO mice.

Collectively, data here and in previous studies (*Diefenbach et al., 2001*; *Guerra et al., 2008*; *Thompson et al., 2017*) revealed multiple roles of RAE-1 molecules on macrophages, tumor cells, endothelial and other cells. It is possible that the frequency or duration of the interaction between NKG2D and RAE-1δ may modulate its functional effect on NK cells, revealing different effects in vitro vs. in vivo. Or, perhaps other signals in the tumor environment influence this outcome. It is also possible that RAE-1δ on TAMs has an as-yet-undiscovered effect on macrophage biology or other features of the tumor microenvironment. Indeed, a previous report documented expression of RAE-1 molecules in microglia in mice with experimental allergic encephalitis, and RAE-1 mRNA expression was found to correlate with CSF-1-induced microglial proliferation in vitro (*Djelloul et al., 2016*). It is also possible that macrophage RAE-1δ interacts with and modulates other NKG2D-expressing cells in tumors. The interactions of immune cells in vivo with NKG2D ligands expressed by distinct untransformed cell types, as well as with transformed and infected cells, appears to impact immune cell activation and function in several respects, in some cases in opposing fashion, and will be the continued subject of future research.

## Materials and methods

### Mice and in vivo procedures

C57BL/6J mice were bred from mice obtained from The Jackson Laboratory (Bar Harbor, ME). RAE-1-KO mice were previously generated in our lab using CRISPR-Cas9 and guide RNAs targeting the open reading frames of the *Raet1d* and *Raet1e* genes, as described (*Deng et al., 2015*). KP mice contain an inducible activating mutation in the proto-oncogene *Kras* and an inducible deletion mutation in the tumor suppressor gene *Trp53* (*DuPage et al., 2009*; *DuPage et al., 2012*) and were bred from mice obtained from The Jackson Laboratory. TRAMP mice contain a transgene expressing the SV40 large T and small T antigens under the rat probasin promoter (*Gingrich et al., 1996*). All mice were maintained at the University of California, Berkeley in accordance with guidelines from the Animal Care and Use Committee. Sex- and age-matched (8- to 12-week-old) mice were used for the experiments.

All transplant tumor models were injected subcutaneously using an insulin syringe (BD Biosciences, San Jose, CA) after suspension in 100 ul PBS. B16-BL6 cells and derivatives were injected at a dose of $1 \times 10^6$ cells, and parental RMA-S cells and derivatives were injected at $5 \times 10^6$ cells per mouse. These high doses were used to establish tumors of uniform size, and tumors were harvested for most analyses at 10–17 days post-injection upon reaching approximately 1 cm in diameter. To generate KP sarcomas, lentivirus-expressing Cre recombinase was generated as described (*DuPage et al., 2009*), and 25,000 PFU was injected intramuscularly into the right hind leg of KP mice. KP sarcomas were harvested when they reached a size of approximately 1 cm in diameter.

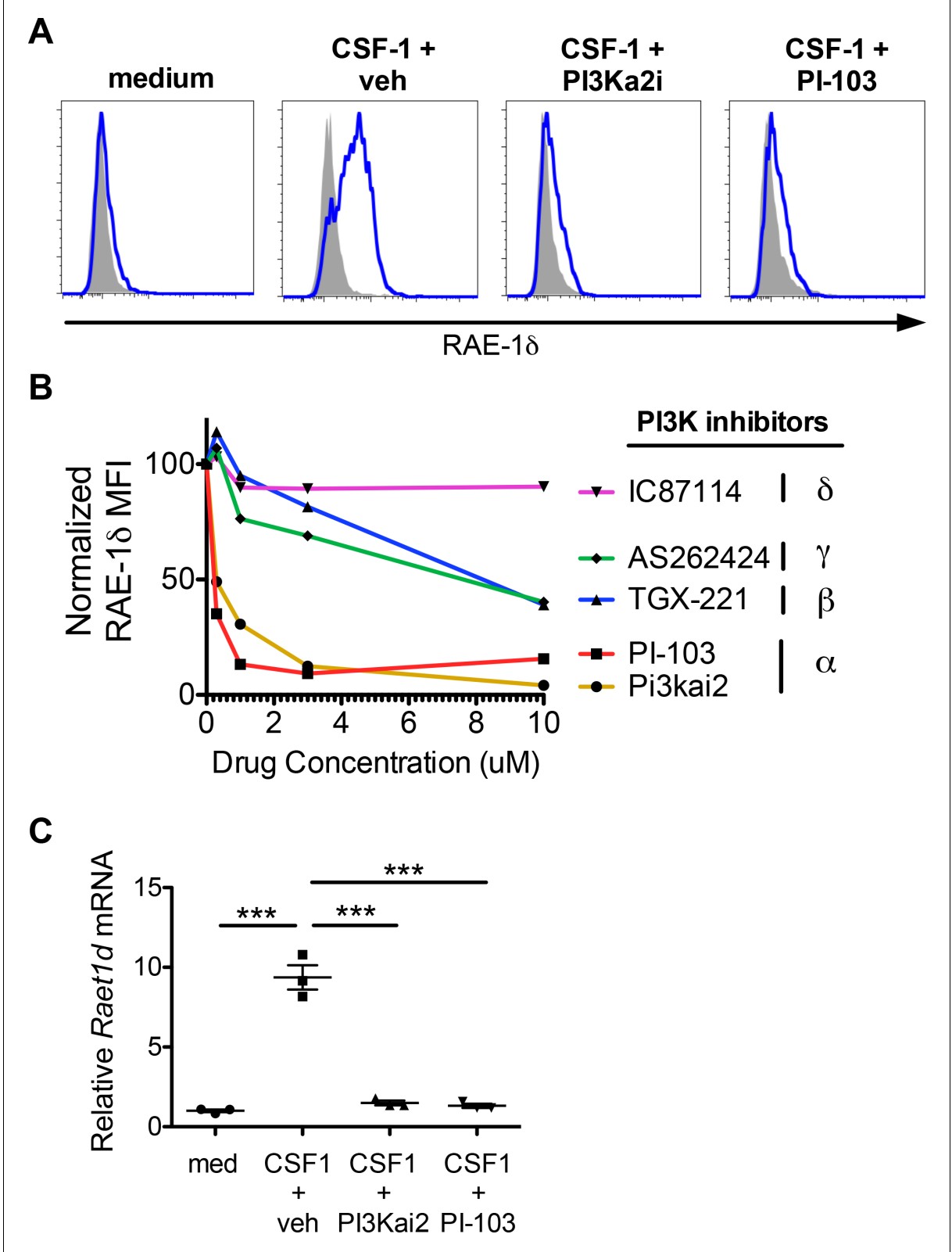

**Figure 6.** PI3Kα signals are required for macrophage RAE-1δ induction by CSF-1. (**A**) Peritoneal wash cells were stimulated with CSF-1 plus vehicle control or PI3Kα inhibitors at 3 µM, and macrophage RAE-1δ was analyzed at 24 hr. (**B**). Relative macrophage RAE-1δ MFI 24 hr after stimulation with CSF-1 plus the indicated concentrations of the indicated PI3K inhibitors. (**C**) Relative *Raet1d* mRNA levels 24 hr after macrophage stimulation with CSF-

*Figure 6 continued on next page*

*Figure 6 continued*

1 plus vehicle control or PI3Kα inhibitors at 3 µM. Statistical significance was determined using one-way ANOVA with Bonferroni post-tests. Data are representative of 3–4 independent experiments.

DOI: https://doi.org/10.7554/eLife.32919.014

The following figure supplement is available for figure 6:

**Figure supplement 1.** Induction of phospho-S6 by CSF-1.

DOI: https://doi.org/10.7554/eLife.32919.015

In some experiments, mice were given blocking antibody (200 ug/injection) against CSF-1 or CSF-1R by i.p. injection using the schedule shown in the figures and legends.

Spleens were dissociated by mashing through a 70 uM filter into PBS. To dissociate tumors for flow cytometry, tumors were excised and minced using a sharp blade, and then incubated in complete medium with 3.5 mg/ml Collagenase D, 1 mg/ml Collagenase IV for 30 min at 37°C with rotation. Cells were then pipetted up and down rigorously 100 times to create a single cell suspension, with additional 10 min, 37°C incubations as needed.

## CSF-1 ELISA

CSF-1 concentrations were analyzed by standard sandwich ELISA. The capture antibody (clone 5A1) was used at 1 ug/ml. Recombinant CSF-1 (Peprotech) was used as a standard. The detection antibody (biotinylated polyclonal anti-CSF-1, R and D systems cat # BAF416) was used at 0.5 ug/ml. Avidin-HRP and TMB substrate (ebioscience) were used for detection. To quantify CSF-1 levels in tumor microenvironments, tumors were dissociated as described above, and the supernatants from the dissociation were subjected to CSF-1 ELISA; intra-tumor concentrations were calculated according to measured tumor volumes calculated using the modified ellipsoid formula: $V = 0.5 \times [(\text{length} + \text{width})/2]^3$, and the volume of dissociation supernatant.

## RNA, cDNA and qPCR

Total RNA was isolated from cells using the RNeasy kit (Qiagen, Hilden, Germany) and converted to cDNA using the iScript system (Bio-Rad, Hercules, CA) according to the manufacturer's instructions. cDNA was subjected to real-time PCR using SsoFast EvaGreen supermix (Bio-Rad) in the presence of primers to amplify *Raet1d* mRNA, or the transcripts of the housekeeping genes β-actin and Rpl19, in a CFX96 RT-qPCR thermocycler (BioRad). Relative mRNA values for *Raet1d* were normalized to the levels of the housekeeping genes, using CFX96 software.

## Cell culture

All cell culture was performed in a humidified 37°C incubator at 5% $CO_2$. Cells were cultured in DMEM or RPMI media (Life Technologies, Carlsbad, CA) supplemented with 5% fetal calf serum (Omega Scientific, Tarzana, CA), 0.2 mg/ml glutamine, 100 U/ml penicillin, 100 µg/ml streptomycin (Sigma–Aldrich, St. Louis, MO), 10 µg/ml gentamicin sulfate (Lonza, Basel, Switzerland), and 20 mM HEPES (Thermo Fisher Scientific, Waltham, MA). Cell lines were obtained from ATCC, authenticated by expression analyses for cell line-specific markers, and routinely tested negative for mycoplasma. For generation of bone marrow-derived macrophages, bone marrow cells were cultured in medium supplemented with 10 ng/ml CSF-1 or GMCSF for 7 days, with fresh medium added every two days.

## Ex vivo peritoneal macrophage stimulation

Cells were obtained by peritoneal lavage of C57BL/6 mice. Briefly, mice were euthanized and injected i.p. with 5 ml ice-cold PBS using a 24-gauge needle, and the peritoneal lavage fluid was then captured using the same syringe. Cells were washed in complete medium and cultured in 12- or 6-well non-TC-treated cell culture plates (Corning, Corning, NY) for 12–48 hr. In some experiments, medium was supplemented with recombinant cytokines (Peprotech, Rocky Hill, NJ) and/or blocking antibodies as indicated. In other experiments, medium was supplemented with conditioned medium from tumor cell lines, which was filtered through a 0.22 uM filter to remove cellular debris. B16-conditioned medium was concentrated 20X using a 10 kDa centrifugal filter unit (cat #UFC901008, Millipore-Sigma). After culture, cells were washed to remove suspension cells, and

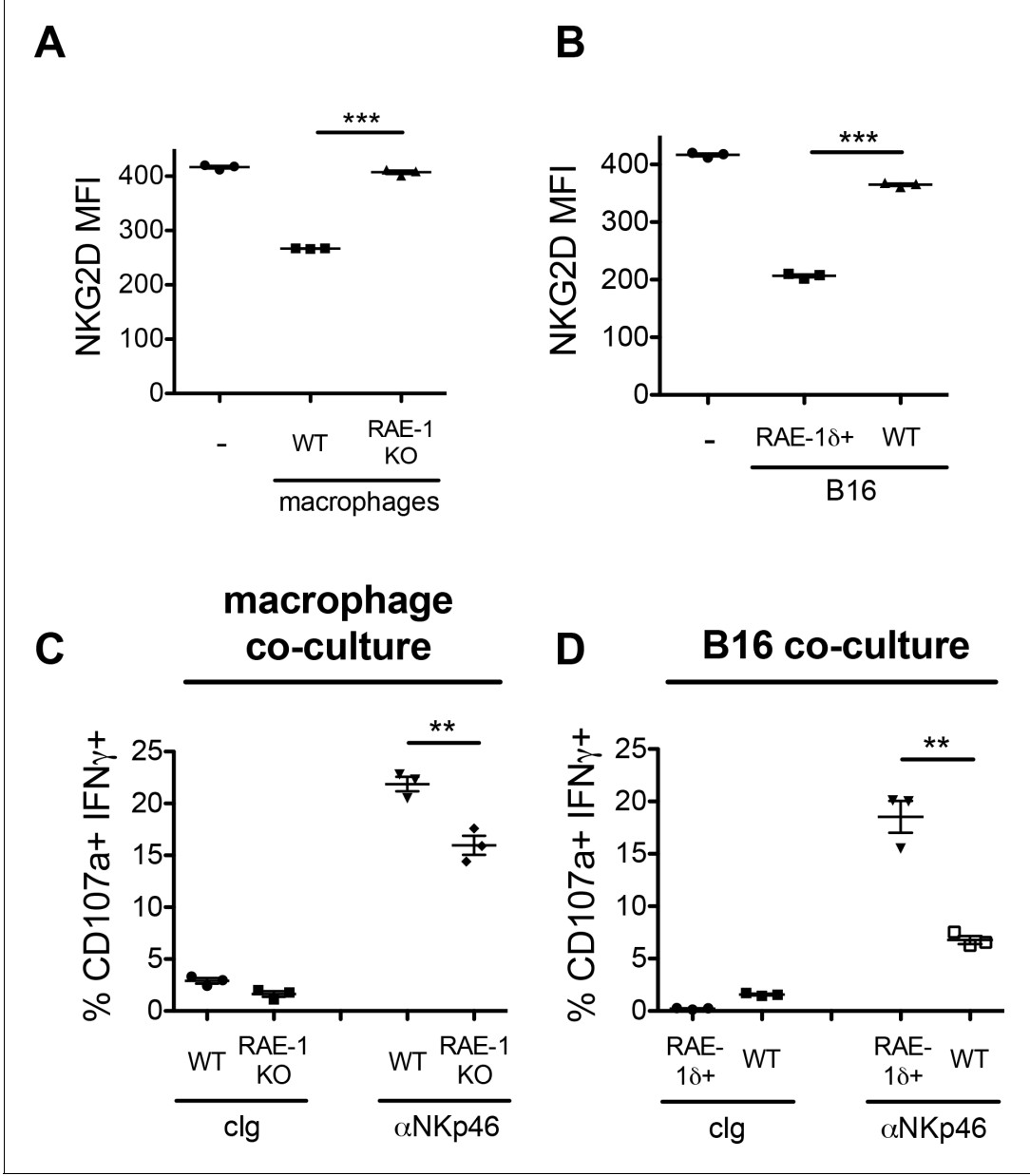

**Figure 7.** Co-culture of NK cells with RAE-1δ-expressing macrophages and tumor cells. (**A**) Peritoneal macrophages from WT or RAE-1-KO mice or were stimulated with 10 ng/ml CSF-1 for 48 hr and then co-cultured with WT splenocytes for 18 hr, and NKG2D levels were analyzed by flow cytometry. (**B**) B16 or B16-RAE-1δ cells were co-cultured with WT splenocytes for 18 hr, and NKG2D levels on NK cells were analyzed by flow cytometry. (**C**) WT splenocytes were co-cultured with CSF-1-stimulated WT or RAE-1-KO macrophages for 18 hr, followed by 5 hr stimulation with plate-bound antibody against the NK cell activating receptor NKp46, or control Ig, and NK cell IFNγ and degranulation were analyzed by flow cytometry. (**D**) WT splenocytes were co-cultured with B16 or B16-RAE-1δ cells for 18 hr, followed by 5 hr stimulation with plate-bound antibody against the NK cell activating receptor NKp46, and NK cell IFNγ and degranulation were analyzed by flow cytometry.

DOI: https://doi.org/10.7554/eLife.32919.016

The following figure supplement is available for figure 7:

**Figure supplement 1.** Tumor associated macrophage numbers and RAE-1δ expression in RAE-1-KO and NKG2D-KO mice.

DOI: https://doi.org/10.7554/eLife.32919.017

macrophages were lifted by vigorous pipetting of ice-cold PBS. Macrophages were identified as live F4/80 + cells by flow cytometry.

## Flow cytometry and FACS

For all flow cytometry experiments, single cell suspensions were generated and incubated for 20 min with supernatant from the 2.4G2 hybridoma to block FcγRII/III receptors, followed by incubation with fluorochrome- or biotin-conjugated specific antibodies for an additional 20 min. In some experiments, an additional incubation with fluorophore-conjugated streptavidin (Biolegend) was performed. For phospho-S6 staining, cells were cultured for the indicated time, and an equal volume of 37°C-prewarmed Cytofix solution (BD Biosciences) was added for 10 min at 37°C. Cells were then suspended in Perm Buffer III (BD Biosciences) for 30 min at 4°C, then washed with regular flow cytometry buffer before staining with anti-phospho-S6 and lineage markers. All flow cytometry samples were analyzed on a LSR Fortessa or LSR Fortessa X20 (BD Biosciences) and data were analyzed with FlowJo software (Tree Star Inc.). Dead cells were excluded from analysis using DAPI (Biolegend) or Live-Dead fixable dead cell stain kits (Molecular Probes) following the manufacturer's instructions.

## Antibodies

We used the following antibodies: from Biolegend: anti-CD3ε (clone 145–2 C11), anti-CD11b (clone M1/70), anti-CD19 (clone 6D5), anti-NKp46 (clone 29A1.4), anti–NK1.1 (clone PK136), anti-Ter119 (clone TER-119), anti-Ly6G (clone 1A8), anti-Ly6C (clone HK1.4), anti-F4/80 (clone BM8) mouse IgG2b isotype control, and rat IgG2b isotype control; from eBioscience: anti-CD45.2 (clone 104), from R and D Systems: anti-RAE-1δ (clone 199205), anti-RAE-1ε (clone 205001), anti-MULT1 (clone 237104), polyclonal anti-H60 (cat # BAF1155); from BioXCell: anti-CSF-1R (clone AFS98), anti-CSF-1 (clone 5A1); from Cell Signaling: anti-phospho-S6 (clone D57.2.2E). For flow cytometry analysis of NKG2D ligands, antibodies were biotinylated in house using the EZ-Link-Sulfo-NHS-LC biotin kit (Thermo Fisher).

## *Csf1* knockout, complementation, and overexpression

Guide RNA sequences targeting the *Csf1* open-reading frame were cloned into the Cas9-expression plasmid px330. Guide RNA sequences are as follows, with bold indicating the PAM: GACGAC-CAGGCGGCCCGCTT**GGG** and ATGGAATCCACGTGCAGGGT**TGG**. B16 cells were co-transfected with both px330 plasmids containing the guide RNAs targeting *Csf1*. Seven days after transfection, cells were single cell cloned. Clones were analyzed by ELISA for CSF-1, and two CSF-1-negative clones were selected for further experiments. To restore CSF-1 expression in B16-*Csf1*-KO cells or express *Csf1* in RMA-S cells, we used an MSCV-IRES-Thy1.1 plasmid containing cDNA encoding secreted CSF-1, a kind gift from Dr. Richard Stanley (Albert Einstein College of Medicine). Thy1.1 +cells were sorted by FACS, and CSF-1 production was confirmed by ELISA.

## NK responsiveness assay

To analyze the responsiveness of NK cells ex vivo, 96-well high-binding flat-bottom plates (Thermo Fisher) were coated overnight with PBS plus anti-NKp46 or control Ig at 5 ug/ml. Plates were washed three times with PBS before stimulation. Cells were cultured in the coated plates for 5 hr in the presence of Golgi-Stop and Golgi-Plug (1:1000 each) (BD Biosciences), 1000 U/ml human IL-2, and fluorophore-conjugated anti-CD107a (0.5 ug/ml) (Biolegend). After stimulation, cells were stained for extracellular markers to identify NK cells and then subjected to intracellular staining for IFN-γ, followed by flow cytometry analysis.

## Statistics and sample size

All statistical analysis was conducted using Prism software (Graphpad, La Jolla, CA), as indicated in the figure legends. Statistical significance is indicated as follows: $*p<0.05$, $**p<0.01$, $***p<0.001$. For most data sets, pilot experiments were performed with a small sample size (usually n = 3) to determine approximate experimental variances and effect magnitudes, and this information was used to determine sample sizes for subsequent experiments.

## Acknowledgements

We are grateful to all members of the Raulet lab, as well as former Raulet lab members for their helpful feedback on this manuscript. We thank Hector Nolla, Alma Valeros, Kartoosh Heydari for

their invaluable help with cell sorting and maintenance of the flow cytometry facility at UC-Berkeley. We thank Dr. Richard Stanley for providing the CSF-1-expression vector. Research reported in this publication was supported by NIH/NCI grants R01-CA093678 (DHR) and F31CA203262 (TWT), and a research grant from Innate Pharma, SAS (DHR). The content is solely the responsibility of the authors and does not necessarily represent the official views of the National Institutes of Health or Innate Pharma.

## Additional information

### Competing interests
David H Raulet: is a co-founder of Dragonfly Therapeutics, and serves on the Scientific Advisory Boards of Innate Pharma, Aduro Biotech and Ignite Immmunotherapy; he has a financial interest in all four companies and received research support from Innate Pharma, and may benefit from commercialization of the results of this research. The other authors declare that no competing interests exist.

### Funding

| Funder | Grant reference number | Author |
|---|---|---|
| National Cancer Institute | R01 CA093678 | David H Raulet |
| Innate Pharma, SAS | | David H Raulet |
| National Cancer Institute | F31 CA203262 | Thornton W Thompson |

The funders had no role in study design, data collection and interpretation, or the decision to submit the work for publication.

### Author contributions

Thornton W Thompson, Conceptualization, Data curation, Formal analysis, Funding acquisition, Validation, Investigation, Visualization, Writing—original draft; Benjamin T Jackson, P Jonathan Li, Alexander Byungsuk Kim, Validation, Investigation, Writing—review and editing; Jiaxi Wang, Kristen Ting Hui Huang, Lily Zhang, Validation, Investigation; David H Raulet, Conceptualization, Resources, Supervision, Funding acquisition, Methodology, Project administration, Writing—review and editing

### Author ORCIDs

Thornton W Thompson http://orcid.org/0000-0002-6352-8238
Alexander Byungsuk Kim http://orcid.org/0000-0002-6425-4566
David H Raulet http://orcid.org/0000-0002-1257-8649

### Ethics
Animal experimentation: This study was performed in strict accordance with the recommendations in the Guide for the Care and Use of Laboratory Animals of the National Institutes of Health. All of the animals were handled according to approved institutional animal care and use committee (IACUC) protocols of the University of California - Berkeley under protocol #AUP-2015-10-8058.

### Decision letter and Author response
Decision letter https://doi.org/10.7554/eLife.32919.021
Author response https://doi.org/10.7554/eLife.32919.022

## Additional files

### Supplementary files
• Transparent reporting form
DOI: https://doi.org/10.7554/eLife.32919.018

## Data availability

All data generated or analysed during this study are included in the manuscript and supporting files

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
