## [Decision Letter]

Thank you for submitting your article "Tumor-derived CSF-1 induces the NKG2D ligand RAE-1δ on tumor-infiltrating macrophages" for consideration by *eLife*. Your article has been reviewed by three peer reviewers, and the evaluation has been overseen by a Reviewing Editor and Michel Nussenzweig as the Senior Editor. The reviewers have opted to remain anonymous.

The reviewers have discussed the reviews with one another and the Reviewing Editor has drafted this decision to help you prepare a revised submission.

Summary:

Three reviewers, experts in NK cell biology, and a Reviewing Editor have evaluated your paper and they found it to be of interest to the readership of *eLife*. In particular, the manuscript convincingly describes a novel mechanism for up-regulating NKG2D ligand expression via CSF-1 produced by tumor cells. However, their opinion was that the paper needed revision before it is acceptable for publication. They discussed the review of your manuscript, allowing them to come to a consensus statement.

Essential revisions:

1) The primary concern for this manuscript is that the functional importance of this finding needs validation. While its importance may be challenging to demonstrate in vivo, an in vitro experiment, as described by reviewer #2, point #4 would nicely address this issue.

2) The Reviewing Editor is aware of another manuscript submitted to *eLife* by the authors. Since, it appears to be relevant to the current manuscript, the authors are advised to submit it as a companion manuscript with the revision. Hopefully, by the time the revision is submitted, more detailed description of the other manuscript can be incorporated into the Discussion.

3) The full reviews from all three reviewers are appended below in order to allow the authors to enhance the manuscript. They also should be addressed in the revised manuscript. In particular, additional information (reviewer #1), additional data (reviewer #2, point #1), and controls (reviewer #2, point #3; reviewer #3, specificity of anti-RAE mAb) are desirable, and should be presented in the revised manuscript. Other clarifications should be addressed as noted. A comprehensive analysis of RAE-1 members in different myeloid cells and tumor models is not required.

Reviewer #1:

The paper by Thompson et al. describes the ability of CSF-1 to specifically stimulate the expression of RAE-1δ on tumor-infiltrating macrophages.

The finding is novel and interesting, but a major weakness is represented by the lack of evidence on the pathophysiological relevance of this finding. In addition as the role of PI3K p110α activation in the regulation of NKG2D ligand expression was already reported, it would be important to provide novel evidence on the down-stream signaling pathway involved in up-regulation of RAE-1δ mRNA expression.

Moreover, a number of experimental details are needed to strengthen the results presented. In particular:

- The authors should provide more details on the tumour models analyzed. It is unclear what they mean with "established tumors": i.e. How many tumor cells were injected, which was the volume of the tumour analyzed, when the tumour was dissociated after transplantation, was RAE-1δ expression on tumor-infiltrating macrophages stable during tumor progression? This last point seems quite relevant as macrophages modify their phenotype during tumour progression.

- Does the in vivo treatment with anti-CSF-1 or anti-CSF1-R antibody affect the number of tumor-infiltrating macrophages? Can the authors show the CSF-1 tumor levels after in vivo anti-CSF-1 antibody administration?

- Is the growth rate of B16, B16-Cfs1 KO, and B16-Cfs1 KO with a restored expression of CFS^-1^, comparable? Were TAM from the different tumors harvested at the same time point? The authors need to provide more details.

Reviewer #2:

In this study Thompson and colleagues describe a novel mechanism for induction of NKG2D ligand tumor-associated macrophages. They show that CSF-1 secreted by tumor cells drives expression of RAE-1δ on macrophages via PI3K dependent signals. This work displays an interesting and novel mechanism used by tumor cells to subvert the NKG2D mediated NK cell response. The findings described are of great interest in how the tumor microenvironment and infiltrating cells can drive immune escape.

1) For Figure 1C it would be helpful to also show pooled data for the noted tumor models, rather than just the representative histogram. If space is an issue the pooled data could be added to the supplementary figures.

2) For consistency in experimental design it would be nice, but not critical, to show anti-CSF-1 treatment on the KP sarcoma model in Figure 4B.

3) A better control for the CSF1-CRISPR KO in Figure 5A and B would be a Cas9 (alone) transfected cells (without guide RNAs) to ensure that the transfection itself is not impairing the ability of the B16 cells to produce CSF1, as well as other possible cytokines and chemokines. That being said, the add back experiment (5C) does seem to indicate that CSF1 is the main mediator of RAE-1δ induction.

4) Given the implications of tumor mediated induction of NKG2D ligands on tumor infiltrating macrophages and the expertise of the Raulet lab it seems logical to evaluate the effect of these CSF-1 induced RAE-1δ expressing macrophages on NK cells. While doing this in vivo would be interesting, evaluation of NKG2D desensitization on NK cells co-cultured with macrophages pre-treated with B16 supernatants or B16 supernatants with CSF-1 blocking antibody would be very interesting and biologically relevant. This would drive home the biologic point of tumor mediated RAE-1δ induction on TAMs.

Reviewer #3:

The manuscript entitled: "Tumor derived CSF-1 induces the NKG2D ligand RAE-1d on tumor infiltrating macrophages" by Thompson et al. reveals a mechanism of regulation of NKG2D-ligands in mice on tumor-infiltrated macrophages. The manuscript is well written, and experiments are well performed, however data are rather descriptive and the functional importance of RAE-1 expression on TAMs is not addressed. Data are cited from a submitted manuscript (Introduction, second paragraph and others) that should be potentially included in the manuscript.

The authors should address the following points to improve the manuscript:

• Introduction: There have been several reports about RAE-1 regulation on myeloid cells. These should be cited and critically discussed. For example:

Djelloul et al., 2016; Nausch et al., 2008.

• Abstract and Introduction: "NKG2D ligands on subset of healthy cells in tumor bearing animals" it is unclear what is meant by healthy cells. Please change accordingly.

• RAE-1 molecules are very similar. Evidence should be provided that the mAb used is specific for RAE-1d and not for other RAE-1s (this should be cited in Materials and methods).

• In Figure 1, tumor associated monocytes are tested –.how were these cells defined? Are these cells similar to MDSC? Gating dot plots for blood monocytes and peritoneal macrophages should be included (Figure 1—figure supplement 1C).

• A comprehensive analysis of different RAE-1 members on different subsets of tumor infiltrating myeloid cells in different tumor models would be very informative.

• Functional analysis of the interaction of RAE-1+ TAMs and NKG2D expressing cells needs to be provided.

---

## [Author Response]

Essential revisions:1) The primary concern for this manuscript is that the functional importance of this finding needs validation. While its importance may be challenging to demonstrate in vivo, an in vitro experiment, as described by reviewer #2, point #4 would nicely address this issue.2) The Reviewing Editor is aware of another manuscript submitted to eLife by the authors. Since, it appears to be relevant to the current manuscript, the authors are advised to submit it as a companion manuscript with the revision. Hopefully, by the time the revision is submitted, more detailed description of the other manuscript can be incorporated into the Discussion.

We thank the reviewers for raising this concern. The recent acceptance of our companion manuscript “Endothelial cells express NKG2D ligands and desensitize anti-tumor NK responses” – PMID 29231815 – to *eLife* allows us to rigorously discuss this issue. We offer revisions of the current manuscript under consideration, along with the following analysis for your consideration. In this response letter, data from the companion manuscript will be referenced as (Endothelial, Figure X) for clarity.

The functional role of host RAE-1 molecules in modulating NK responses is the central focus of the Endothelial manuscript. We clearly show that host RAE-1 molecules engage NKG2D and desensitize NK cells at steady-state and in tumors (Endothelial, Figures 1, 2, and 5). Importantly, we find that radio-resistant cells are the dominant source of RAE-1 molecules responsible for NKG2D engagement and NK desensitization at steady state and in tumors, whereas radiosensitive cells have a small or negligible effect (Endothelial, Figures, 4 and 5). Macrophages in the tumor models studied are completely radio-sensitive and therefore are replaced with donor cells in chimeras (unpublished data available upon request). Using antibody blockade, we showed that RAE-1ε is the relevant RAE-1 molecule responsible for NKG2D engagement and NK desensitization, whereas RAE-1δ had little to no effect (Endothelial, Figures 1 and 5). We went on to show that endothelial cells in lymphoid tissue, but not any other non-hematopoietic cells, express RAE-1ε at steady state (Endothelial, Figure 4). Furthermore, RAE-1ε was super-induced on endothelial cells within all tumors tested (Endothelial, Figure 5). These data are consistent with a model in which endogenous expression of RAE-1ε by endothelial cells engages NKG2D and causes NK cell desensitization, mitigating anti-tumor NK responses.

The present manuscript under consideration describes parallel experiments concerning regulation of RAE-1δ (but not RAE-1ε) induction on tumor-associated macrophages. We chose not to include functional analyses in the initial draft of this manuscript because the in vivo effect of RAE-1 molecules on NK cell function seemed limited to RAE-1ε (not RAE-1δ) molecules on nonhematopoietic cells, as described in the Endothelial paper. Upon consideration of the reviewers’ helpful comments, we now include data from in vitro experiments that test the role of CSF-1 induced macrophage RAE-1δ on NK function in a cell culture system; we have also added data on the number of intra-tumor macrophages in WT vs. RAE-1-KO mice. These experiments have been added to the revised manuscript (Figure 7) and are discussed in the Discussion section.

For the in vitro functional experiments, NK cells were incubated with CSF-1-stimulated macrophages from WT or RAE-1-KO mice, and subsequently analyzed for NKG2D downregulation or NK responsiveness to plate-bound antibody stimulation. NK cells were also incubated with parental B16 or RAE-1δ-overexpressing B16 cells (this was an attempt to determine if the effect of macrophage-expressed RAE-1δ differed from that of tumor-expressed RAE-1δ). In this in vitro model, incubation with RAE-1δ-expressing macrophages or RAE-1δ-expressing B16 tumor cells both caused efficient downregulation of NKG2D on NK cells and increased NK responses to subsequent stimulation compared to culture with RAE-1-deficient cells (Figure 7). This acute priming effect is similar to findings from previous studies showing heightened NK responses following co-culture with RAE-1-expressing MDSCs (Nausch, Galani, et al., 2008).

In addition, we considered the hypothesis that NK cells were killing RAE-1δ-expressing macrophages in tumors, which should result in fewer TAMs or selection for TAMs with lower RAE1δ expression. However, the number of intra-tumoral macrophages was similar in WT and RAE-1KO mice, and the expression of RAE-1δ on macrophages was similar in WT vs. NKG2D-KO mice. These data have been added to the manuscript (Figure 7—figure supplement 1) and do not support the hypothesis that RAE-1δ-expressing macrophages are targeted for killing in vivo.

Because the in vitro data showed a stimulatory effect of macrophage RAE-1δ (Macrophage, Figure 7), whereas in vivo experiments showed a desensitizing effect of RAE-1ε molecules on endothelial cells (Endothelial, Figures 1, 2, 4, and 5), we now include a robust discussion of these datasets and their implications (Discussion, last paragraph), as suggested by the reviewers. We hope that these revisions address the reviewers’ concerns, and we look forward to future studies to further elucidate the complex role of these ligands and receptors.

3) The full reviews from all three reviewers are appended below in order to allow the authors to enhance the manuscript. They also should be addressed in the revised manuscript. In particular, additional information (reviewer #1), additional data (reviewer #2, point #1), and controls (reviewer #2, point #3; reviewer #3, specificity of anti-RAE mAb) are desirable, and should be presented in the revised manuscript. Other clarifications should be addressed as noted. A comprehensive analysis of RAE-1 members in different myeloid cells and tumor models is not required.

Listed below are point-by-point responses to the remaining reviewers’ comments.

Reviewer #1:The paper by Thompson et al. describes the ability of CSF-1 to specifically stimulate the expression of RAE-1δ on tumor-infiltrating macrophages.The finding is novel and interesting, but a major weakness is represented by the lack of evidence on the pathophysiological relevance of this finding. In addition as the role of PI3K p110α activation in the regulation of NKG2D ligand expression was already reported, it would be important to provide novel evidence on the down-stream signaling pathway involved in up-regulation of RAE-1δ mRNA expression.

We appreciate the reviewer’s suggestion to include data on the physiological role of macrophage RAE-1δ expression. We have attempted to address these concerns, as outlined above in the responses to Essential Revisions 1 and 2. In addition, we acknowledge that our data on the role of PI3K p110α corroborate a previous finding in CMV-infected cells (Tokoyama, Lorin, et al., 2011), and we feel that this common mechanism is an important finding because the regulation of RAE-1 molecules is mediated by diverse signaling pathways in different pathophysiological contexts (see Raulet, Gasser et al., 2014). Unfortunately, we do not have any additional data on the mediators downstream of PI3K; because of the intricacy of this regulatory system and the ongoing nature of these investigations in our lab and others, we respectfully request that such datasets be reserved for future publications.

Moreover, a number of experimental details are needed to strengthen the results presented. In particular:- The authors should provide more details on the tumour models analyzed. It is unclear what they mean with "established tumors": i.e. How many tumor cells were injected, which was the volume of the tumour analyzed, when the tumour was dissociated after transplantation, was RAE-1δ expression on tumor-infiltrating macrophages stable during tumor progression? This last point seems quite relevant as macrophages modify their phenotype during tumour progression.

We thank the reviewer for this helpful feedback. We have clarified the details of our analysis in the Results and Materials and Methods sections. For all experiments, we injected high doses of tumor cells (1 x 10^6^ B16 and 5 x 10^6^ RMA-S) in an attempt to standardize tumor size. Tumors were typically harvested upon reaching approximately 1 cm in diameter, roughly 10-17 days after injection. According to the reviewer’s suggestion, we have included time course data on macrophage RAE-1δ expression in B16 tumors. Mice were injected with 1 x 10^6^ cells, and tumors were harvested at early (day 6), mid (day 11) and late (day 16) time points. Tumor-infiltrating macrophages did not show significant differences across these time points. These data have been added to the manuscript (Figure 1—figure supplement 2).

- Does the in vivo treatment with anti-CSF-1 or anti-CSF1-R antibody affect the number of tumor-infiltrating macrophages? Can the authors show the CSF-1 tumor levels after in vivo anti-CSF-1 antibody administration?

As shown in Figure 4—figure supplement 1A, intra-tumoral macrophage numbers are reduced upon anti-CSF-1R treatment as early as day 5 post-treatment, whereas macrophage RAE-1δ expression is reduced as early as day 2 post-treatment. Because cells of the macrophage/monocyte lineage are known to depend on CSF-1 for survival, these results are expected. The anti-CSF-1 antibody (5A1) is a well-established blocking antibody, but because of the technical difficulties of quantifying CSF-1 in vivo after blocking antibody injection, we do not have data on the intra-tumoral levels of CSF-1 following blockade of the cytokine. The finding that the antibody has large effects (reducing RAE-1δ expression and eventually depleting macrophages) shows that the availability of the antibody is clearly sufficient to have strong biological effects.

- Is the growth rate of B16, B16-Cfs1 KO, and B16-Cfs1 KO with a restored expression of CFS^-1^, comparable? Were TAM from the different tumors harvested at the same time point? The authors need to provide more details.

We thank the reviewer for this comment. Similar to the comment addressed above, mice were given high-dose tumor cells in an effort to standardize tumor growth. At the high dose of 1 x 10^6^ cells, B16, B16-CSF1-KO and B16-CSF-1-add-back grew at similar rates (data presented in Author response image 1) and were harvested at the same time point (14 +/- 3 days post-injection) for analysis. We have added these details to the manuscript (Results, subsection “Tumor-derived CSF-1 is required for RAE-1δ expression by TAMs in vivo”).

**Author response image 1. respfig1:** WT mice were injected with 1 x 10^6^ of the indicated tumors. Volumes were analyzed at day of harvest (day 15).

Reviewer #2:In this study Thompson and colleagues describe a novel mechanism for induction of NKG2D ligand tumor-associated macrophages. They show that CSF-1 secreted by tumor cells drives expression of RAE-1δ on macrophages via PI3K dependent signals. This work displays an interesting and novel mechanism used by tumor cells to subvert the NKG2D mediated NK cell response. The findings described are of great interest in how the tumor microenvironment and infiltrating cells can drive immune escape.1) For Figure 1C it would be helpful to also show pooled data for the noted tumor models, rather than just the representative histogram. If space is an issue the pooled data could be added to the supplementary figures.

We thank the reviewer for this useful suggestion. We have added this pooled data to Figure 1—figure supplement 2C. Note that this represents compiled data from multiple experiments because we did not have a single experiment where we simultaneously analyzed RAE1δ in the different tumors.

2) For consistency in experimental design it would be nice, but not critical, to show anti-CSF-1 treatment on the KP sarcoma model in Figure 4B.

We agree that having this data would be good for matching the experimental setup in the B16 tumors, but unfortunately because of the lengthy tumorigenesis time for KP mice and their use in unrelated experiments, we do not have this data, and we respectfully request that the manuscript be allowed to move forward without it.

3) A better control for the CSF1-CRISPR KO in Figure 5A and B would be a Cas9 (alone) transfected cells (without guide RNAs) to ensure that the transfection itself is not impairing the ability of the B16 cells to produce CSF1, as well as other possible cytokines and chemokines. That being said, the add back experiment (5C) does seem to indicate that CSF1 is the main mediator of RAE-1δ induction.

We agree that the add back experiment represents a strong control that tumor-derived CSF1 mediates TAM RAE-1δ induction. Relatedly, we point to our use of two different CSF1-KO B16 clones (Figure 5A and 5B) as an additional control for potential off-target effects of Cas9. We believe that these data, taken together, make a compelling case that the defect is due to CSF1 deficiency and not other features of the cells. Although we did not test Cas9 transfection alone, we know that in general, transfection of B16 cells does not affect their ability to make CSF-1 (data not shown).

*4) Given the implications of tumor mediated induction of NKG2D ligands on tumor infiltrating macrophages and the expertise of the Raulet lab it seems logical to evaluate the effect of these CSF-1 induced RAE-1δ expressing macrophages on NK cells. While doing this* in vivo *would be interesting, evaluation of NKG2D desensitization on NK cells co-cultured with macrophages pre-treated with B16 supernatants or B16 supernatants with CSF-1 blocking antibody would be very interesting and biologically relevant. This would drive home the biologic point of tumor mediated RAE-1δ induction on TAMs.*

We thank the reviewer for this comment, and we agree that additional information on the functional role of RAE-1 would be useful. We hope that our experiments described above in response to Essential Revisions 1 and 2 are satisfactory.

Reviewer #3:The manuscript entitled: "Tumor derived CSF-1 induces the NKG2D ligand RAE-1d on tumor infiltrating macrophages" by Thompson et al. reveals a mechanism of regulation of NKG2D-ligands in mice on tumor-infiltrated macrophages. The manuscript is well written, and experiments are well performed, however data are rather descriptive and the functional importance of RAE-1 expression on TAMs is not addressed. Data are cited from a submitted manuscript (Introduction, second paragraph and others) that should be potentially included in the manuscript.The authors should address the following points to improve the manuscript:• Introduction: There have been several reports about RAE-1 regulation on myeloid cells. These should be cited and critically discussed. For example:Djelloul et al., 2016; Nausch et al., 2008.

We thank the reviewer for these helpful additions; the findings from these papers are now described in the revised manuscript in the Discussion section.

• Abstract and Introduction: "NKG2D ligands on subset of healthy cells in tumor bearing animals" it is unclear what is meant by healthy cells. Please change accordingly.

Based on the context of the text, we have revised this to more explicitly denote the cells referenced.

• RAE-1 molecules are very similar. Evidence should be provided that the mAb used is specific for RAE-1d and not for other RAE-1s (this should be cited in Materials and methods).

We thank the reviewer for this comment. We analyzed the specificity of the RAE-1 antibodies by staining B16 cells transduced to specifically express RAE-1δ or RAE-1ε. The antibodies showed high specificity for their targets. The anti-RAE-1δ did show some staining of the RAE-1ε – expressing cells, but the difference in staining intensity was nearly 100-fold. These data have been added to Figure 1—figure supplement 2A. Furthermore, we showed in the Endothelial paper that the RAE-1δ antibody blocked binding of NKG2D-Fc to RAE-1δ and had no effect in blocking binding of NKG2D-Fc to RAE-1ε – and vice-versa (Endothelial Figure 1—figure supplement 1A).

• In Figure 1, tumor associated monocytes are tested – how were these cells defined? Are these cells similar to MDSC? Gating dot plots for blood monocytes and peritoneal macrophages should be included (Figure 1—figure supplement 1C).

Monocytes in tumors were identified as CD45+ CD11b+ Ly6G-neg F4/80-low Ly6C-hi. In blood, monocytes were identified as by CD3e-neg CD19-neg Ly6G-neg Ly6C-hi. In the peritoneum, macrophages were identified clearly as the F4/80-hi cells. In accordance with the reviewer’s comment, we have added these gating strategies to the manuscript (Figure 1—figure supplement3).